# Extracellular Vesicles in Modifying the Effects of Ionizing Radiation

**DOI:** 10.3390/ijms20225527

**Published:** 2019-11-06

**Authors:** Tünde Szatmári, Rita Hargitai, Géza Sáfrány, Katalin Lumniczky

**Affiliations:** Department of Radiation Medicine, Division of Radiobiology and Radiohygiene, National Public Health Center, 1221 Budapest, Hungary; hargitai.rita@osski.hu (R.H.); safrany.geza@osski.hu (G.S.); katalin.lumniczky@osski.hu (K.L.)

**Keywords:** extracellular vesicles, nanocarriers, ionizing radiation, intercellular signaling

## Abstract

Extracellular vesicles (EVs) are membrane-coated nanovesicles actively secreted by almost all cell types. EVs can travel long distances within the body, being finally taken up by the target cells, transferring information from one cell to another, thus influencing their behavior. The cargo of EVs comprises of nucleic acids, lipids, and proteins derived from the cell of origin, thereby it is cell-type specific; moreover, it differs between diseased and normal cells. Several studies have shown that EVs have a role in tumor formation and prognosis. It was also demonstrated that ionizing radiation can alter the cargo of EVs. EVs, in turn can modulate radiation responses and they play a role in radiation-induced bystander effects. Due to their biocompatibility and selective targeting, EVs are suitable nanocarrier candidates of drugs in various diseases, including cancer. Furthermore, the cargo of EVs can be engineered, and in this way they can be designed to carry certain genes or even drugs, similar to synthetic nanoparticles. In this review, we describe the biological characteristics of EVs, focusing on the recent efforts to use EVs as nanocarriers in oncology, the effects of EVs in radiation therapy, highlighting the possibilities to use EVs as nanocarriers to modulate radiation effects in clinical applications.

## 1. Introduction

Radiotherapy is one of the essential treatment modalities for cancer, applied alone or in combination with chemotherapy or other treatment modalities. According to statistics, approximately 50% of cancer patients receive radiotherapy [1]. The major obstacle of radiotherapy, causing the failure of treatment and often the recurrence and metastasis of the tumor, is the radioresistance of cancer cells. Consequently, great effort has been made to study the causes and mechanisms of radioresistance, to find modalities to overcome radiotherapy tolerance of cancer cells and to increase radioresistance of normal cells in the tumor microenvironment.

The extracellular environment of multicellular organisms contains various mobile membrane-coated structures, called extracellular vesicles (EVs) [2]. EVs have a diameter of 50–5000 nm, and they are actively excreted by cells. Emerging evidence supports that active release of EVs into the extracellular environment is a universal cellular process [2,3,4]. EV release is amplified by stress responses, including response to ionizing radiation (IR) [5,6]. EVs can circulate in body fluids throughout the organism and transport different molecules originating from parent cells. This horizontal transfer of various nucleic acids (microRNAs (miRNA), short interfering RNAs (siRNA), mRNAs, long noncoding RNAs (lncRNA), DNAs), proteins, receptors, enzymes, and lipids to specific recipient cells to activate downstream signaling pathways and, thus, influence the cellular metabolic state, physiology, and function supposed to be the most important role of EVs [7,8,9,10,11]. EVs can regulate gene expression through the novel translation of delivered mRNAs and post-translational regulation through miRNAs [7].

Therefore, as natural carriers, EVs are important mediators of intercellular communication at short and long distances [2,4,12] regulating a broad range of physiological cellular processes in both normal and diseased states, including tumor development. Cell signaling pathways are affected by the delivery of different RNA species to target cells via EVs. Small RNAs could be suitable therapeutics, but they are difficult to be delivered in the target cell, because they are very prone to RNA degradation in the extracellular space. Moreover, crossing the plasma membrane is also difficult since they are negatively charged and have higher molecular weight. Hence, when packed into EVs and, thus, protected by a lipid bilayer, RNAs are more efficiently transported to the target. It was also demonstrated that EVs may act as antigen-presenting vehicles to stimulate immune responses and lead to activation of T-lymphocytes [13,14]. On the other hand, tumor cells and cells in tumor microenvironments secrete EVs that may contribute to tumor progression by promoting angiogenesis and tumor cell migration in metastasis [15,16,17,18]. Moreover, tumor-derived EVs may have immunosuppressive effects, inhibiting cytotoxic activity of NK cells, suppressing proliferation of NK-cells and T-lymphocytes, and blocking T-cell directed apoptosis [19,20,21]. EVs may also modulate the susceptibility/infectability of the recipient cell to viruses and prions [3]. 

On the other hand, EVs have the ability to protect against intracellular stress [22,23,24], thus, they may be utilized for therapeutic purposes. Moreover they can be engineered to carry certain therapeutic drugs or RNAs, miRNAs, siRNAs. Having similar size as other synthetic nanocarriers, but being able to avoid degradation and escape recognition by the body’s immune system, they have the potential to be used as nanocarriers for modulating radiation effects. 

For the use of EVs as nanocarriers, first we need to understand the interaction between EVs and cells, both in terms of EV release and uptake. In the first part of the paper we review the current knowledge about EV formation, release as well as uptake and internalization in the receptor cells. We also review the methods of EV engineering and the currently used modalities of EVs as nanocarriers. The role of EVs in chemotherapy resistance was extensively studied; the effects exercised by EVs on radioresistance are much less investigated. This review aims to summarize the functions of EVs with an emphasis on radiotherapy-associated features and the possibilities to use EVs as radiation modifiers.

## 2. Biological Characteristics of EVs

### 2.1. EV Types and Biogenesis 

EVs are complex structures composed of a phospholipid bilayer with membrane proteins, carrying soluble cytosolic components of the donor cell.

EVs can be divided into three main groups (exosomes, microvesicles, and apoptotic bodies) on the basis of their cellular mechanisms of generation and size distribution (Figure 1). The different types of vesicles are present at the same time in the extracellular environment. 

Exosomes were first described by Trams et al. [25]. They are released both by healthy cells and by tumor cells, and are found in abundance in blood, saliva, urine, and breast milk [2,26]. Exosomes have a diameter of 50–100 nm, overlapping the size range of viruses [2,4]. They are formed inside the early endosomes by inward budding of the membrane, thus generating intraluminal vesicles [26,27]. Multivesicular bodies (MVBs) containing intraluminal vesicles marked with tetraspanins or lysosomal-associated membrane proteins fuse with the plasma membrane and secrete the vesicles from the cell in a controlled manner [14,28]. These exocytosed vesicles are called exosomes. Exosome markers include proteins involved in their endosomal biogenesis, such as Alix, tumor susceptibility gene 101 protein (TSG101), tetraspanins (CD63, CD81, CD9), and lysosomal-associated membrane proteins (LAMP1 and LAMP2) [29,30,31].

Microvesicles (MVs) have a diameter of 50–1000 nm, overlapping the size range of bacteria. They are generated by budding/blebbing of the cell membrane [2,4,14]. MVs may contain proteins, mRNA, miRNA, membrane receptors, and even infectious agents (viruses, prions) [3].

Finally, apoptotic bodies were first described by Kerr (1972) [32]. Apoptotic bodies have a diameter of 1–5 μm, they are released as blebs of cells undergoing apoptosis, and they may contain organelles, fragmented DNA, and oncogenes [33,34].

Since the currently available EV isolation methods are not fully suitable to discriminate between these categories [4], and the core properties such as size, morphology, composition and markers are actually overlapping [35], it is probable that studies declaring to investigate either exosomes or MVs analyzed instead a mixture of EV types. Therefore, the current guidelines of EV studies suggest “extracellular vesicle” as a generic name for all these vesicles naturally released from cells and delimited by a lipid bilayer [36], unless their MVB origin can be clearly demonstrated [37].

### 2.2. EV Composition 

The composition of EVs primarily depends on the type and maturation state of the donor cell [38] but environmental stressors could also influence it. The cargo of EVs includes mRNA, miRNA, other small noncoding RNAs (e.g., tRNA, siRNA) and long noncoding RNA [7,39,40], genomic DNA fragments [41], proteins, and lipids (Figure 1).

EVs have been demonstrated to selectively incorporate RNA [42]. For instance, some miRNA may be specifically loaded into EVs by a regulated process, as their levels in EVs are higher than those in donor cells [7]. It was demonstrated that transferred mRNA can be translated to new proteins in the recipient cells [7], and transferred miRNA is able to modulate gene expression [38,42], indicating that they are functional in their new location.

EV composition is highly complex with hundreds to thousands of various proteins on both the outside and inside of EVs. Exosomes carry protein families associated with their endosomal origin, such as Alix and TSG101 [43]. EVs are also enriched in membrane proteins that are known to cluster into microdomains at plasma membranes or at endosomes, such as tetraspanins (e.g., CD63, CD81, CD82, CD53, CD37) [29,44]. Moreover, they contain proteins associated with lipid rafts, ’conserved’ proteins (heat shock proteins, cytoskeleton proteins, metabolic proteins, MHC proteins), and cell-type specific proteins [31,45,46].

Regarding lipid species, the amounts of several lipids, such as phosphatidylinositol, phosphatidylethanolamine, sphingomyelin, and phosphatidylserine were found to be higher in EVs than in donor cells [47]. 

### 2.3. EV Uptake 

For the delivery of their cargo, EVs must first bind to the target cell surface and then internalize either by fusing directly with the plasma membrane or with the endosomal membrane following endocytosis and delivering the cargo to the cytoplasm [38,48]. Some studies suggest that EVs can be incorporated uniformly by every cell type [49], but there is much evidence for the selective targeting of specific cells [50,51,52]. When compared the internalization efficiency of a brain tumor cell line and astrocyte derived EVs in tumor cells, the uptake of astrocyte-derived EVs was significantly lower than that of tumor-derived EVs [53]. Similarly, mantle-cell lymphoma (MCL) derived EVs were selectively taken up by MCL cells and less by T-cell leukemia and bone marrow stroma cells [54]. Target cell specificity is probably determined by adhesion molecules, such as tetraspanins and ECM proteins (e.g., laminin and fibronectin) [55,56]. It was demonstrated that only minor changes in exosomal tetraspanin complexes strongly influenced target cell selection in vitro and in vivo [52]. These small transmembrane proteins on the EV surface, with a role in cell adhesion and migration may help target the EVs to certain cell types. Several tetraspanins—CD63, CD9, and CD81—are the most well-known EV markers [57]. In addition to tetraspanins, integrins [58] and proteoglycans [59,60] from the surface of EVs are the key molecules for selective binding, resulting in targeted biological effects. Tetraspanins interact with integrins on recipient cell membranes and promote EV-cell docking [52,61,62]. For example, Tspan8-CD49d complexes were described in the EV uptake by rat aortic endothelial cells [63]. EVs with Tspan8-CD49d complex on their surface were internalized by endothelial and pancreatic cells through the ligand ICAM-1 (CD54) [52]. Hoshino et al. demonstrated that different exosomal integrins were associated with metastasis at different organs: exosomal integrins α_6_β_4_ and α_6_β_1_ with lung metastasis, while exosomal integrin α_v_β_5_ to liver metastasis. Inhibition of these integrins decreased exosome uptake and metastasis [58]. Interaction with tetraspanins is required for viruses to enter the cell [64] and it is proposed that EV-cell binding, EV uptake and targeting could occur through similar processes [52,57]. 

EV uptake by recipient cells can be very rapid, as EVs were identified inside cells from 15 min after initial introduction to phagocytic cells [65]. The mechanisms that mediate the uptake of EV cargo to target cells seem to be highly variable [48,61]. A wide range of evidence suggests that EV (both exosome and MV) uptake is typically an energy-dependent process that can occur through more than one mechanism and that requires a functioning cytoskeleton of the recipient cell [48,61]. Intraluminal vesicles are transported by the cytoskeleton and fuse with the endosomal membrane where they deposit their cargo [48]. Although the energy-dependent endocytic process appears to be the main mechanism for EV uptake, passive membrane fusion is also a possible entry route [38,66,67]. The mechanism by which EVs are internalized may vary among different cell types, and may influences their biological effects on recipient cells [65]. 

The endocytic routes used by EVs for internalization are diverse, such as clathrin-dependent endocytosis, caveolin-mediated uptake, phagocytosis, macropinocytosis, and lipid-raft mediated internalization [48,65,68] (Figure 2). Clathrin-dependent endocytosis induces membrane curvature around the EV developing a clathrin-coated vesicular bud, which is released to the cytosol through membrane scission, then it undergoes a clathrin un-coating in the cytosol and finally fuses with the endosome [69,70]. Caveolae are small, cave-like invaginations in the cell membrane, rich in cholesterol, sphingolipids, and caveolins, which can be internalized into the cell [71,72]. During phagocytosis, the cell cytoskeleton is rearranged to create a cup-shaped invagination around the EV, which is then internalized via membrane scission creating an endosome [65,73]. During macropinocytosis, the cell rearranges its cytoskeleton to generate plasma membrane ruffles that fold back on themselves around the EV, forming a lumen of a macropinosome [74,75]. Lipid rafts are membrane microdomains rich in protein receptors, sphingolipids, and cholesterol [71]. EVs may also exert their function on cells by direct interaction between membrane molecules of EVs and receptors of the target cells [56] or by vesicle-cell membrane fusion to deliver EV cargo into the cytosol of the target cell [38]. High level of lipid rafts in EV membrane may facilitate their fusion with the plasma membrane [76].

It is not known yet what the decisive mechanism for the EVs to “choose” an internalization route is, but it seems likely that a heterogeneous population of EVs enters into a cell via more than one route. The used internalization route appears to depend both on the type and origin of EVs, and target cell type [51,77]. Determining which pathway the internalization will follow and whether these processes result in cargo delivery and functional changes would be essential to use the EVs as nanovehicles. A suitable approach may be to first follow the delivery of a cargo ubiquitously present in all type of EVs to determine under which conditions the physiological effect takes place [37]. 

## 3. EVs and Radiation

The interplay between EVs and IR was less studied as compared to the impact of EVs on cancer cells in general. Nevertheless, in the last years a number of studies emerged regarding both the impact of irradiation on EVs’ composition and function [78,79,80,81,82] and the effect of EVs on the behavior of cells after IR [83].

### 3.1. Role of EVs in Radiation-Induced Bystander Effects

The conventional model in radiobiology states that the effects of IR on living organisms are caused by direct damage to a cellular target, particularly DNA, as a result of direct absorption of radiation energy or by indirect effects of reactive oxygen species (ROS). ROS are produced from radiolysis of water and cause molecular damages by chemical reactions, due to their unpaired electrons. The radiation-induced DNA damages can be various base modifications, single or double strand breaks, DNA–DNA and DNA–protein cross-links, and different chromosome aberrations [84]. This concept has changed over time, since many studies proved that not only can cells directly hit by radiation beam be damaged, but there are similar changes in the neighboring “bystander” cells, in distant cells or in the progenies of irradiated cells [85]. As opposed to the previously described “targeted” effects, these are called non-targeted effects of IR. When non-targeted effects of IR occur in the progeny of irradiated cells, they are called genomic instability, and when they occur in non-irradiated neighboring or distant cells, they are called local or systemic bystander effects [86,87,88]. 

Radiation-induced bystander effects (RIBE) consists of radiation induced adaptive responses [89], low-dose hypersensitivity [90], damage in DNA such as micronuclei formation, mutations, sister chromatid exchanges, modified gene expression, alteration in the miRNA profile, induced oxidative stress, and cell death [10,11,91,92,93,94,95]. RIBE emerge in non-irradiated cells that have received damaging signals from directly irradiated cells via intercellular communication [96,97,98,99]. Activated DNA damage response pathways (e.g., p53) are required for irradiated cells to secrete bystander effectors [100]. It is likely that multiple pathways are involved in the RIBE. Intercellular communication can be mediated through cell-cell contact (gap junction) [101] or transfer of secreted soluble molecules. Soluble transmitting factors could be cytokines, including interleukins [93,102], TGF-β [103], TNF-α [94,104], nitric oxide (NO) [103,105], Ca fluxes [106] ROS [96], and miRNA [107,108]. 

There is growing evidence that radiation-induced genomic instability and bystander effects are partially mediated by EVs [8,9,10,11,109]. For instance, it was shown that nonirradiated cells showing a bystander effect could themselves induce bystander effects in other naïve cells through EVs released [110]. In this mode of intercellular communication, signal molecules packed in EVs are prevented from dilution and degradation by extracellular enzymes in the extracellular environment.

Increasing evidence supports that EV release is elevated in a dose-dependent manner after IR through activation of stress-inducible pathways of EV secretion [17,110]. This process was shown to be stimulated by DNA damage activated P53-transcription factor [100,111].

However, data on radiation-induced changes in EV content is limited [9,10,17,91]. It was demonstrated that EV mediated miRNA (e.g., miR-21, miR-34c) transfer plays important role in RIBE [9,10,11], and proteins could also be important [110]. 

### 3.2. Role of EVs as Natural Nanocarriers in Radio- and Chemotherapy 

Multiple studies showed that EVs from different tissues have a role in modulating radio- and chemotherapy responses of tumors. Most of the studies found that native, unmodified EVs can enhance resistance of tumor cells to radiotherapy or reverse radiation injuries, more rarely do they have sensitizing effects. EVs mediate radiation resistance by inhibiting apoptosis [81,112], interfering with cell cycle regulation, delivering proteins that increase tumor cell survival or inducing DNA repair [113]. Furthermore, they can induce the generation of cancer stem cells through epithelial-mesenchymal transition (EMT) and have important roles in remodeling the microenvironment by the tumor cells, mediating hypoxic injury or adaptation to hypoxia [114] (see Figure 3).

EVs may promote tumor cell survival following irradiation by induction of DNA repair mechanisms. In an in vitro model of head and neck cancer, production of EVs was elevated by irradiation, and these EVs through miRNA transfer induced accelerated DNA repair, thus enhancing radioresistance [115,116]. EVs from irradiated breast cancer cells were taken up by human primary mammary epithelial cells, inducing an increased phosphorylation of ATM, Histone H2AX, and checkpoint kinase 1 (Chk1) in recipient cells indicating the induction of DNA damage repair responses [117]. In another study, both murine and human MSC derived-EVs were able to ameliorate radiation damage to murine bone marrow cells by stopping the radiation induced growth inhibition, DNA damage, and apoptosis [118]. 

In donor cells, EVs can mediate therapy resistance by decreasing intracellular drug concentrations [81,119] or, similarly, by reducing intracellular levels of tumor suppressive miRNAs. Furthermore, EVs can inhibit pro-apoptotic signaling by sequestering and/or removing the pro-apoptotic proteins or miRNAs from the cells. It was found that EVs from cells transfected with caspase-3 contained higher levels of caspase-3 as compared to the donor cells. Moreover, these vesicles were taken up by untransfected cells, but these cells did not undergo apoptosis [120]. In colorectal cancer, miR-145/−34a withdrawal from cancer cell-derived EVs increased 5-fluoruracil resistance of these cells by decreasing apoptosis [121].

In recipient cells EVs can increase intracellular levels of certain miRNAs and proteins with role in radiation or chemotherapy response. Apoptosis inhibition can be initiated by the surface receptors carried by the EVs which can activate anti-apoptotic pathways [119]. IL-6 or CD41 (integrin α-IIb) transferred by EVs inhibited apoptosis of tumor cells [119,122,123]. Furthermore, EVs can interfere with several other anti-apoptotic signaling pathways, such as p38, p53, JNK, Raf/MEK/ERK, mTOR and PI3k- Akt [81,124,125] to induce radio- and chemotherapy resistance. Another mechanism by which EVs can confer resistance to therapy-sensitive tumor cells is by transmitting transcription factors and miRNAs that alter cell cycle control. MiR-222, transferred by EVs from drug-resistant cancer cells to drug-sensitive cell, transferred the docetaxel resistance to originally drug-sensitive cells as well. The authors have shown that miR-222 from EVs downregulated PTEN pathway, inhibiting cell cycle arrest [126,127]. 

EVs can further increase radiation resistance by promoting cell migration causing cancer cells to leave the irradiated area. It was demonstrated that EVs derived from irradiated squamous head and neck carcinoma cells conferred a migratory phenotype to recipient cells through enhancement of Akt pathway [78]. Similarly, in glioblastoma cells radiation affected the molecular composition of EVs towards a migratory phenotype [17].

If tumor cells enter in dormancy - which is characteristic to cancer stem cells [128]—they can escape the damaging effects of irradiation, since they have a slow rate of cell cycling and IR can kill mainly the proliferating cells. Cancer stem cells are the most radioresistant cells within a tumor, whereas non-stem cancer cells are more radiosensitive. EVs were shown to be able to transfer this therapy resistance from resistant to sensitive cells, through miRNAs [127]. EVs can induce a cancer stem cell-like phenotype and dormancy in tumor cells [129,130]. This was demonstrated with EVs originating from bone marrow niche, which triggered dormancy of metastatic breast cancer cells [130]. EVs isolated from conditioned medium of carcinoma-associated fibroblast promoted clonogenicity and tumor growth of cancer stem cells upon treatment with 5-fluorouracil or oxaliplatin [131]. In diffuse large B cell lymphoma model, EVs induced a cancer stem cell (CSC) like phenotype and dormancy through WNT pathway, associated with doxorubicin resistance [132]. Breast cancer fibroblast derived EVs induced a CSC like phenotype in breast cancer cells, associated with radiochemotherapy resistance [73]. In another study, EVs induced de-differentiation of lung carcinoma cells to a more CSC-like phenotype and reduced cell cycle progression, leading to methotrexate resistance [74,75]. 

The first steps of radiation damage are radiation-induced energy deposition, then generation of ROS. EVs were shown to contribute to both generation of ROS and protection against them. EVs can deliver NOX2 from activated macrophages to injured neurons to confer ROS generation, mediating axon outgrowth [133]. On the other hand, MSCs and MSC derived EVs have been reported to inhibit ROS and reduce oxidative stress through secretion of ROS scavengers such as superoxide-dismutase and protect injured cells against ROS by transporting anti-inflammatory cytokines [134]. MSC-derived EVs can mediate repair of radiation-induced bone marrow stem cell injury. Quesenberry et al. found that MSC-EVs injected intravenously following 500 cGy irradiation of mice, leads to the recovery of peripheral blood counts and restoration of the engraftment of bone marrow, and inhibition of irradiation-induced gene expression in peripheral blood and bone marrow. They also demonstrated that both murine- and human-derived vesicles are effective against radiation-induced injury and that a mixture of exosomes and MVs has superior effects compared to either exosomes or MV alone [135,136].

In the process of tumor formation, due to high proliferation rate, extensive hypoxic regions develop in the tumor. The microenvironment of hypoxic tumor cells confers a more aggressive cancer phenotype and worse prognosis. Radiotherapy is also less effective on hypoxic cells, because, beside the direct effects of radiation, the DNA damaging effect of IR takes place mainly via ROS, and in low oxygen environment, less ROS are produced, and, consequently, DNA damage is reduced [137]. One explanation of this radiosensitizing effect of oxygen is the oxygen fixation hypothesis. According to this hypothesis DNA lesions that are produced by IR can be repaired under hypoxia but are fixed with the chemical participation of molecular oxygen. Nevertheless, this hypothesis does not take into consideration the role of enzymatic DNA repair [138]. The quantitative measurement of the oxygen effect is the oxygen enhancement ratio, the ratio of doses under hypoxic vs. aerated conditions necessary to produce the same level of cell killing.

In the regulation of the hypoxia-adaptation mechanisms, EV-mediated crosstalk between cancer cells and stroma is very important [114]. The effects of tumor or stroma derived EVs on tumor cells within hypoxic microenvironment in regulating different features of cancer have been extensively investigated [139,140,141,142,143,144,145,146,147]. It was demonstrated that under hypoxic conditions, tumor cell-derived EVs are actively released into the tumor microenvironment and by transporting different regulatory miRNAs and signaling proteins, have essential roles in tumor growth and invasiveness, angiogenesis, drug- and radiation resistance, cancer stemness, and metastasis [114,148]. Jung et al. showed that EVs released by hypoxic cancer cells are preferentially taken up by hypoxic cancer cells and these EVs could be engineered to carry anticancer drugs. Moreover, uptake of engineered EVs caused increased apoptosis and slower tumor growth in vivo [149]. In this context, EVs could be ideal nanocarriers to deliver radiosensitizers for hypoxic cancers.

In order to reverse the radiation resistance mediated by the miRNAs in the EV cargo, it is possible to knockdown certain miRNAs via EVs. MiRNA inhibitors can be loaded directly in EVs similar to miRNA loading [150]. MiR-21 was inhibited in macrophage cell-derived EVs by loading a miRNA-inhibitor in EVs, and resulted in regulation of gastric cancer cell proliferation [151]. Similarly, anti-miR oligonucleotides can be loaded successfully in EVs in order to inactivate a specific miRNA. The delivery of anti-miR-9 to the resistant glioblastoma multiforme cells through EVs sensitized the cancer cells to Temozolomide, as shown by increased cell death and caspase activity [152].

EVs with radiosensitizing properties can be obtained by selection of donor cells. It is known that EVs derived from MSCs, can have radiosensitizing effects. In a melanoma mouse model combination of RT and MSC derived EVs had therapeutical benefit, with a control of tumor growth and metastases [153]

Another possibility is to enhance radiosensitivity of tumor cells is by reprogramming CSCs via EVs. Using adipose derived stem cell EVs with osteoinductive potential, it was possible to induce osteogenic differentiation of CSCs, with the loss of CSC –like phenotype and properties, consequently becoming more radiosensitive [154].

## 4. Why are EVs Good Nanocarriers?

The ideal nanocarriers as drug delivery agents must avoid degradation, escape recognition by the body’s immune defenses, have reduced clearance rates, high cellular uptake, should show targeted delivery of loaded therapeutics and should have controlled release of cargo molecules upon selective stimuli [155]. 

Based on several characteristics such as their nanoscale dimension, biocompatibility and the ability to target the tissues, EVs have been proposed for a long time as natural nanocarriers. Examples of currently used nanocarriers include liposomes, polymeric nanoparticles, micelles, carbon nanotubes, gold nanoparticles, solid lipid nanoparticles, and dendrimers, but all have major drawbacks, such as poor biocompatibility, limited intrinsic targeting ability, poor cellular uptake, and tissue toxicity [155] (Table 1). Out of these nanocarriers, only liposomes and polymeric nanoparticles were used in clinical trials [156]. Even for liposomes, toxicity, lack of stability, and ability to evade the host immune system remain a concern [157]. Polymeric nanoparticles seem more stable than liposomes but their biocompatibility is still poor [156].

EVs have the potential to be unique nanocarrier system, because they possess most of the properties of being a good delivery vehicle. EVs naturally transfer their cargo to recipient cells [158], they possess a high stability: they are difficult to be degraded, due to the protective lipid bilayer membrane, and due to their small size, thus they can travel through the body without being degraded [159]. EV membrane structure is similar to that of cells [160], but compared to the plasma membrane of the cell, EV membranes are enriched in cholesterol, sphingomyelin, annexin, phosphatidylserine, and glycosphingolipids [161], their internal cargo being protected. They are present in all biofluids, released from all different cell types in the body. They show high biocompatibility because of their natural origin, so they do not activate the immune system. 

Moreover, EVs do not accumulate in different tissues for long-term, causing low systemic toxicity. Due to their small dimensions, they can even cross tissue barriers, such as the blood-brain barrier [159,162]. It was hypothesized that EVs travel by transcytosis through endothelial cells, entering the endothelial cells via the endocytic system and leave them through multivesicular bodies [163,164].

EVs have enhanced cellular uptake compared to several synthetic drug delivery systems because of their natural ligand-receptor and internalization system: they have key proteins on their surface such as tetraspanins and integrins which will determine the rate of uptake. The cellular targeting and the specificity are also facilitated by certain chemokines, antigen recognition and certain membrane molecules (e.g., integrins) which selectively attach to certain cell types making the cargo delivery specific for designated cell-types.

EVs, as they have a native cargo, can be loaded by various methods with different drugs with high loading efficiency (see later). Although cells naturally release just a limited amount of EVs, the large scale, clinical-grade production of EVs is possible, for example high-density cell culture in bioreactors were applied in drug industry [165,166]. 

The application of EVs as nanocarriers has also its limitations: first of all, EV isolation and procession are not standardized yet [36], and second, the mechanisms of EV release and uptake are still insufficiently described. EV detection following uptake is also technically challenging and although diverse methods exist (reviewed by [171]) they all have advantages and disadvantages (Table 2). Moreover, in some cases when EVs were used as nanocarriers, activation of the immune system was observed. Furthermore, long-term EV stability and in vivo EV toxicity have been scarcely investigated and to date, only a few clinical trials have been performed [172].

EV-based nanocarrier systems can be used in two different modalities, based on their cargo: natural EVs, when the EV is used to carry different molecules as it is secreted by cells, and EVs modified by externally loading different bioactive molecules into them.

## 5. Cargo Loading in EVs 

Bioengineered or “artificial” EVs are constructed for different purposes: to increase traceability, targetability or to introduce therapeutic cargo inside EVs. 

EVs can be exogenously loaded with nucleic acids such as small RNAs and DNA such as plasmids. Moreover, they can be loaded with water-soluble drugs, due to their hydrophilic core [180]. 

The methods used for loading of EVs with therapeutic cargo can be classified into three categories: (1) engineering parental cells with DNA encoding therapeutically active compounds or miRNA, siRNA oligonucleotides which are then released in EVs; (2) loading parental cells with a foreign material, e.g., a drug, which is then incorporated into EVs; and (3) direct modifications of EVs previously isolated from parental cells ex vitro.

(1) Indirect EV modification via bioengineering of parental cell: with this approach, parental cells are genetically modified prior to EV isolation. As a result, isolated EVs will carry the therapeutic drug/molecule produced by the engineered cells. The genetic modifications aim to introduce genes of therapeutic proteins, or plasmids containing the therapeutic gene in the cell. With this approach Haney et al. transfected macrophages ex vivo with a plasmid DNA encoding catalase, an antioxidant enzyme. Cargo of EVs secreted by these cells contained increased amount of catalase mRNA, pDNA and active catalase. Moreover, they demonstrated that EVs efficiently transferred their content to target cells resulting in de novo protein synthesis [181]. 

Loading of therapeutic RNA cargo in EVs is also possible by transfecting oligonucleotides of interest (mRNAs, [182], miRNAs [183,184,185] siRNAs [162]) directly into parent cells [186]. Transfection of precursor miRNA oligonucleotides in mesenchymal stem cells also resulted in loading of miRNAs into EVs [187,188]. These oligonucleotides will be delivered by EVs into target cells inducing (mRNA) or reducing (miRNA, siRNA) transgene protein expression. This way degradation of different RNAs by RNAses is avoided [182,189]. 

Another purpose of indirect EV engineering is the modification of cell-surface tetraspanins, in order to control targeting, because only minor differences in the tetraspanin panel have a strong influence on target cell selection [52]. Several groups used this approach to introduce tracking agents such as fluorescent proteins or luciferase reporters in tetraspanins [190,191,192]. Stickney et al. constructed a set of fluorescent reporters at selected sites of tetraspanin CD63 for both the inner and outer surface on exosomes, allowing stable integration and exosomal display of the fluorescent protein via gene transfection. Their system was capable of continuous production, secretion, and uptake of EVs [192]. Similarly, EVs can be targeted towards certain cell types by inserting different molecules on their surface that bind to receptors on particular (e.g., cancer) cells; consequently, they can selectively accumulate in the target sites. In addition to the characteristic EV tetraspanins (CD63, CD9, CD81), lactadherin (C1C2 domain) [193,194], lysosome-associated membrane glycoprotein 2b (Lamp-2b) [162], platelet-derived growth-factor receptors (PDGFRs) [182] were successfully engineered. Cells were transfected with a fusion cDNA, consisting of the gene of the chicken egg ovalbumin, OVA (cargo) and the gene of lactadherin C1C2 domain, a protein localized in membranes and secreted in association with EVs, resulting in the loading of EVs with OVA proteins [194].

(2) As a second approach, parental cells can be treated with therapeutic agents before EV isolation, which then are packaged in EVs. A variety of chemotherapeutic drugs were loaded in EVs using this strategy, ex. Paclitaxel, using MSC [195] or Hep2G [196] cells, etoposide, irinotecan, epirubicin, and mitoxantrone [196] using Hep2G cells, doxorubicin, gentamicin, 5-fluorouracil using macrophages [197], or carboplatin using Hep2G cells [196] or macrophages [197]. With this approach, to overcome the relatively low loading efficiency, macrophages are frequently used as parent cells, because they actively engulf basically any type of foreign particles. Several studies applied this approach; in particular, iron-oxide nanoparticles have been loaded into cells together with different therapeutic agents. As a result, the uptake of the drug by the cells as well as the uptake of EVs by recipient cells are enhanced and kinetically modulated and spatially controlled under magnetic field [198,199,200]. Another strategy to increase loading efficiency is to increase concentration of the material and incubation time [191] or to use liposomes as delivery systems since they easily fuse with cell membranes [201]. As mentioned above, loading parental cells with the therapeutic material has the drawback that the incorporation has low efficiency, only a small percentage of the material packed in the cell will be loaded into the EVs. From this point of view, direct modifications of EVs ex vivo can achieve higher efficiencies.

(3) An alternative strategy is to directly load the therapeutic cargo in isolated, purified EVs. Within ex vivo loading of EVs we can distinguish passive and active loading methods. Passive loading strategies rely on spontaneous interactions between the EV and the cargo, meaning that the loaded materials are co-incubated with EVs [202]. Generally, lipophilic small molecules, such as curcumin [203] and the anticancer drugs doxorubicin [204] or paclitaxel [205] were loaded successfully into EVs with this method. The main advantage of the passive loading methods is that they are relatively simple and do not require the addition of other substances into the system. The main drawback of these methods is the low loading capacity.

Active loading methods involve membrane permeabilization strategies. For example, electroporation of isolated EVs is commonly used to transiently permeabilize the EV membrane to enhance the uptake of exogenous miRNA [182,206], siRNA [162,207], and other small molecule compounds and drugs [202]. Transient permeabilization of EV membrane with saponin treatment was also proposed for loading of exogenous material directly into EVs [208,209]. Saponin interacts with cholesterol, generating new pores in the membrane, and the drugs can enter through these pores into EVs. With saponin permeabilization, one concern is the in vivo hemolytic activity of saponin [210]. Alternatively, sonication was used as well for encapsulating materials into EVs. With this method, the EV membrane is sheared by a sonicator in order to create new pores were the drugs can diffuse into EVs. As an example, catalase, which is a larger protein, was loaded in EVs using several different methods such as incubation at room temperature, freeze/thaw cycles, sonication, extrusion, or permeabilization with saponin, and sonication resulted in the highest loading efficiency [209]. 

Each strategy has its advantages and limitations and the efficiency of the EV based drug delivery depends on a variety of factors. When making a choice, one should take in consideration the type of therapeutic cargo, characteristics of both donor and recipient cell types, and conditions suitable for a specific type of EV cargo [211]. Cell-based EV loading strategies typically package only a small fraction of their content into the EVs, making the loading efficiency very low. In contrast, direct loading of ex vivo EVs makes possible that most part of the modified content enters the vesicle. While the limitation of passive loading approaches is the relatively low loading capacity, the limitation of active loading strategies is that the EV membrane can be disrupted during the procedures, compromising the integrity and functionality of them. Moreover, direct loading into EVs is more efficient and uniform method, as it is possible to pool EVs from different isolations, resulting in a high amount of EVs, which then can be loaded with the therapeutic cargo. On the other hand, EVs already carry numerous proteins and nucleic acids, which lower the loading capacity. The success of this method requires also a complete understanding of structural characteristics of EVs. 

## 6. Potential Role of EVs as Drug Delivery Nanovehicles in Radiotherapy

Engineered EVs have been used for targeted drug delivery and gene therapy in a variety of studies thus have the possibility to become radiosensitizing agents as well. Similar to the loading of EVs with any therapeutic cargo, radiosenzitising effects may be achieved by loading three different cargo types: (1) small molecular formulation drugs; (2) radiosensitizing proteins or other natural molecules; or (3) various small RNA specimens such as siRNAs or miRNAs (Figure 4). 

(1) Classical cancer radiosensitizers, including oxygen mimetics and hypoxia-specific cytotoxins are small-molecule formulation drugs. Oxygen is the prototype of radiosensitizers, acts by damage fixation, using its two unpaired electrons to produce new free radicals, initiating a chain reaction [212]. Oxygen mimetics are small biomolecules with similar characteristics, usually compounds with a nitro group with free electrons (nitroimidazoles: misonidazole, etanidazole, pimonidazole, nimorazole, etc.). Hypoxia-specific drugs are bioreductive agents which are selectively toxic to hypoxic cells. These pre-drugs are metabolized into toxic compounds under hypoxic conditions, damaging hypoxic cells. Examples of hypoxia-specific drugs are Tirapazamine, SN30000, and AQ4N [212,213,214,215,216]. All these small molecules are potentially excellent radiosensitizers, but their major shortcomings are their poor tumor penetration, lack of specificity and dose-limiting toxicity in case of oxygen mimetics [212], while for hypoxic radiosensitizers, although preclinical studies were promising, early clinical trials showed either too high toxicity, or no survival advantage [217]. Due to their small size, it might be possible loading these compounds in EVs with the aim of increasing specificity and stability and with this, reducing toxicity on non-target tissues.

Traditional chemotherapeutic drugs, such as paclitaxel and doxorubicin have been successfully loaded into EVs several times [195,204,218,219]. Etoposide, irinotecan, epirubicin, mitoxantrone, gentamicin and 5-fluorouracil delivery through EVs are also under investigation [196,197]. Many of these drugs have been shown to have radiosensitizing properties as well [220], constituting the standard treatment regimens together with radiotherapy for many solid tumors. Although, to the best of our knowledge, their radiosensitizing effects as EV-delivered drugs have not been evaluated yet, they might have therapeutic benefits. Paclitaxel, docetaxel, and doxorubicin were already evaluated as nanoparticle delivered radiosenzitizers, but using different nanoparticle formulation [221,222,223]. Their delivery through EVs might further increase target specificity and the stability of the compounds, EVs being able to avoid degradation and recognition by immune system, as opposed to the synthetic nanoparticles used in these studies.

(2) One example of small molecules loaded in EVs as therapeutic cargo is curcumin, frequently used against a variety of diseases, but mainly as an anti-inflammatory agent [203,224,225]. On the other hand, curcumin is a very interesting molecule in terms of radiobiology as well: it was frequently reported to act as a radiosensitizer for various cancers (pediatric, lymphoma, sarcoma, prostate, gynecologic, pancreas, liver, colorectal, breast, lung, head and neck, and glioma), reviewed by [226,227]. Furthermore, there are several data showing that curcumin has radioprotector effects on normal tissues [226]. It was hypothesized that curcumin exerts its radioprotective effects by reducing oxidative stress and inhibiting inflammatory responses, whereas the radiosensitizing activity might be due to the upregulation of genes responsible for cell death [227] and suppression of NFkB [226]. Consequently, it is possible that delivery of curcumin through EVs to normal and/or tumor cells could be a successful approach for a radiosensitizing purpose. 

STAT3, redox-sensitive transcriptional factor is a proven key mediator of radioresistance [228]. As protein cargo, STAT3 inhibitors were also successfully loaded into EVs [229] and selectively delivered to target cells, inducing apoptosis. Therefore it might be possible that the use of STAT3 inhibitors via EV delivery might show radiosensitizing effects in tumors.

(3) MiRNAs and siRNAs are small, ~22 nucleotide-long RNAs that have been used as a very effective target-specific gene silencing tool for various diseases. MiRNAs are noncoding RNAs found in living organisms with a role in post-transcriptional regulation, which exert their function by complementary pairing with mRNA molecules and silencing them. SiRNAs are double-stranded RNA molecules. Their role is similar to that of miRNAs, the silencing of the expression of specific genes with complementary nucleotide sequences, by binding and degrading mRNA after transcription. Both miRNA and siRNA loading in EVs was successfully applied by various methods for a variety of diseases as described above. Therefore, loading radiosensitizer miRNAs in EVs might be considered, as the use of “naked” miRNAs is limited because of their lack of stability in body fluids, the inability to cross biological barriers which would be compensated by EV delivery. SiRNAs and miRNAs constitute an emerging therapeutic tool in modulating radiation response, since they can modulate a broad range of signaling pathways governing radiation response, such as DNA repair, histone modifications, cell cycle checkpoint control, ROS formation and the antioxidant defense system and other signaling pathways [230]. There are plenty of data showing the radiosensitizing potential of different miRNAs. We will shortly review the most recent findings in this field. These data suggest that miRNAs would be useful radiosensitizers when loading in EVs.

Inhibition of RAD51, a key molecule in DNA repair has been used for radiosensitizing tumor cells [231]. Potentially this effect can be achieved with a higher efficiency through EV-loaded inhibitors: in a study, inhibition of RAD51 was achieved through siRNA against RAD51 loaded in EVs [232]. EV delivery of the siRNA into target glioma cells resulted in RAD51 gene silencing and reproductive cell death of recipient cancer cells [232]. 

Another approach for radiosensitizing is targeting elements of DDR pathways. Ataxia-telangiectasia mutated (ATM) kinase, a key signaling protein in DDR, regulating cell cycle checkpoint activity and DNA repair was shown to be a target of various miRNAs in diverse tumor types, such as miR-421 in neuroblastoma [233], miR-101 [234], and miR-30 [235] in lung cancer. Overexpressing miR-182 targeted BRCA1, an important gene for homologous recombination, inhibiting DNA repair and radiosensitizing tumor cells [236]. MiR-101 exerts its radiosensitizing effects by targeting both ATM and DNA-PK, which are key factors in non-homologous end joining repair [234]. miR-890 and miR-744-3p were shown to inhibit DDR and repair in prostate cancer [237]. P53, the key factor in DNA-damage checkpoint activation and apoptosis can be targeted by miR-125b, miR-504, and miR-33 [230]. Furthermore, miR-30a and miR-205 modulate radiosensitivity of prostate cancer cells by TP53INP1 [238]. Cdc25a, another important checkpoint signaling kinase is a direct target of miR-21 and let-7 [230], and their overexpression leads to enhanced radiosensitivity. MiR-339-5p was shown to enhance radiosensitivity in multiple ways, by targeting Cdc25A in head and neck cancer [239] and by inducing G0/G1 arrest and apoptosis in lung cancer [240]. Histone modification and chromatin remodeling required in DDR also can be targeted by specific miRNAs. MiR-328 sensitizes cells to radiotherapy by targeting histone H2AX both in osteosarcoma [241] and lung cancer [242]; furthermore, in lung cancer mir-138 [243] and miR-30a [235] were also found as radiosensitizing agents via H2AX targeting. 

Furthermore, miRNAs can induce radiosensitivity by targeting other pathways, independent of DDR. Mir-122 induced radiosenitization of lung cancer cells by inhibiting stress response [244]. Interestingly, up-regulation of miR-122 in breast cancer cells with acquired radioresistance promoted cell survival. Authors suggested that miR-122 differentially controlled radiosensitivity by a dual function as a tumor suppressor or oncomiR depending on cell phenotype [245]. Proteins regulating the formation of ROS were also shown to be regulated by miRNAs. MiR-21 downregulates the scavenging of ROS, by targeting SOD2 and SOD3, members of the superoxide dismutase (SOD) family, resulting in increased radiosensitivity. The mitogen activated protein kinase (MAPK) pathway and PI3K/Akt/mTOR pathways have a central role in radiotherapy. Using inhibitors of these pathways have been for long tried as radiosensitizers. MiRNAs seem to be suitable tools in this approach as well. It was demonstrated in several studies that let-7 has radiosensitizing effects acting by suppressing MAPK pathway through KRAS [230]. Mir-875 enhances radiosensitivity by suppressing EGFR in prostate cancer [246], miR-200c in lung cancer cell by targeting VEGFR2 [247] and in breast cancer cells by increasing apoptosis and DSB [248]. 

All these miRNAs represent potential radiosensitizing agents, but their targeted delivery still has major drawbacks. In body fluids RNAs are unstable, due to the rapid degradation by RNAses. Thus, the main challenge remains their delivery to the target tissue without being degraded. Other shortcomings of systemic delivery of miRNAs are particle aggregation, the high percentages of liver toxicities reported, stimulation of immune responses, uptake by macrophages, inefficient uptake by target cells and inefficient endosomal release [249]. To overcome these barriers, several nanovehicles are under investigation. Adding EVs to these systems might lead to development of a good carrier system for radiosensitizing miRNAs.

## 7. Concluding Remarks

EVs are widely recognized as efficient mediators of intercellular communication. Emerging studies deal with the use of EVs as safe and efficient drug and gene delivery vectors for many diseases. It has been recognized that EVs possess superior characteristics as drug delivery systems, such as their natural origin, ability to target specific tissues and long-term stability. For these reasons, we suggest that, while the use of artificial nanomaterials and nanosystems, such as gold nanoparticles as radiosensitizing agents, is already under investigation, the use of EVs as naturally-occurring nanocarriers for radiosensitizing agents has also vast therapeutic potential. Most importantly, miRNAs are very efficient modulators of radiation response, with a potential to be radiosensitizing agents for tumor cells at the clinical level, if a safe and efficient delivery system would be available. On the other hand, the miRNA content of unmodified EVs might be considered a druggable target for radiosensitization purposes as well, miRNAs mediating radiation resistance could be knocked down by inhibitors. 

Since the use of EVs as nanocarriers is still in its infancy, although in the last years significant progress has been made, there are still several drawbacks and shortcomings such as the lack of consensus on the best EV isolation method and the most efficient loading method, the still insufficient targeting, and the lack of toxicity data in the absence of in vivo studies and clinical trials. One major drawback which has to be solved might be the fact that EVs, as cell-derived vesicles, carry their own cargo originating from the donor cell and, depending on their origin and context, some EV content (especially miRNAs) could mediate radioresistance to the cells. Furthermore, it is known that miRNAs are highly related to various diseases, including cancer. Moreover, the same miRNA could be beneficial in some health condition and detrimental in others [250]. For this reason, how exactly the release of these miRNAs from engineered EVs would influence tumor behavior, including its response to radiation, need to be cautiously explored. Thus, future research is needed to solve this contradiction. One possibility might be to choose the donor cell carefully as EV characteristics and contents reflect the cell of origin. For example, with the selection of a suitable model in which the miRNA causing the opposite effect is not naturally expressed [251,252], the unwanted effects might be controlled. From this point of view, the cargo loading method into EVs which involves the loading of parental cells with non-native materials involving “hijacking” cellular biosynthesis to favor the production of specific endogenous material with the desired radiosensitizing properties might be favorable. By supplementing cell culture medium with foreign metabolites, this approach allows these metabolites to be incorporated and produced in the cells, at the expense of the original metabolites. Another possibility for counterbalancing the not desired effects of EV native cargo is to knockdown an unwanted miRNA—in case the EV composition is well-known—for example, by transducing the cells with anti-miRNA oligonucleotides [152].

Another major task is to fully elucidate the cargo packaging mechanisms of EVs. Several cargo sorting mechanisms have been identified for EVs, but, to date, the process is not completely understood. Currently, the lack of methods to interfere with this cargo sorting also hampers their use. An effective selective mechanism especially for EV miRNA sorting system would greatly improve the possibilities to use EVs as carrier systems.

Taken together applying EVs as agents for modifying the biological effects of radiation could be a challenging, but intriguing, attempt that requires further research. The use of EVs for delivery systems in radiobiology requires full understanding of the nature of the donor cells, EV composition, cargo sorting mechanisms, and consent in methods of purification and loading. It is crucial to carefully study the biological characteristics of EVs and to weigh their benefits and disadvantages for modifying radiation responses. Thus, we can hypothesize that with continuous efforts to overcome these issues, EV-mediated delivery of radiosensitizing drugs, miRNAs and other biotherapeutics may result in a selective reduction of radiation resistance of tumor cells, and radiotherapy for cancer patients would become more efficient.

## Figures and Tables

**Figure 1 ijms-20-05527-f001:**
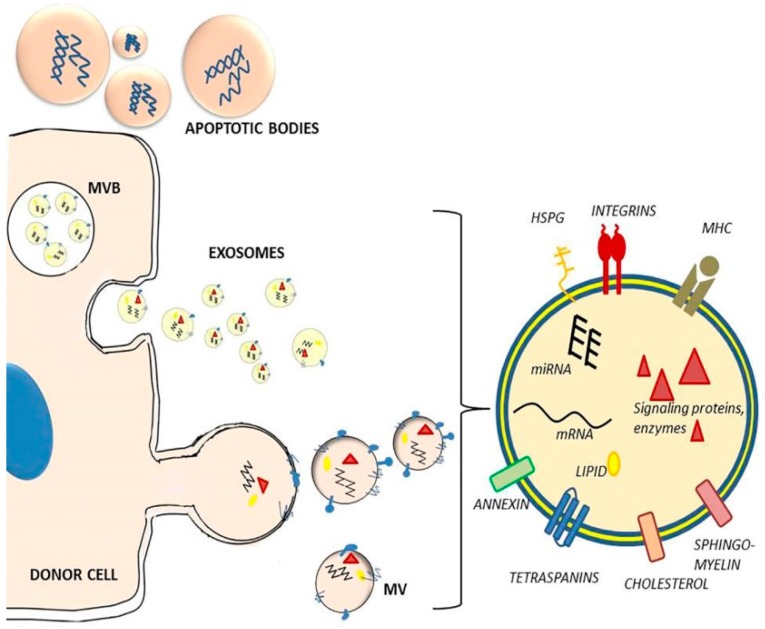
Types, release, and composition of extracellular vesicles. Based on their generation and size distribution EVs can be divided into exosomes, microvesicles, and apoptotic bodies. Exosomes are formed inside endosomes by inward budding of the membrane, generating multivesicular bodies (MVB). Exosomes are released by fusion of MVB with the plasma membrane. Microvesicles (MV) arise as a result of direct budding and fission of the plasma membrane from the cells. Apoptotic bodies are formed during apoptosis, from outward blebbing of the cell surface. EVs are composed of a phospholipid bilayer with membrane proteins (immuno-regulatory molecules such as MHCI, MHCII, integrins, tetraspanins, receptors, heparan-sulfate proteoglycans (HSPG), annexins, cholesterol, sphingolipids, ceramides), carrying soluble cytosolic components of the donor cell, such as miRNAs, mRNAs, signaling proteins, cytoskeletal proteins, enzymes, and lipids.

**Figure 2 ijms-20-05527-f002:**
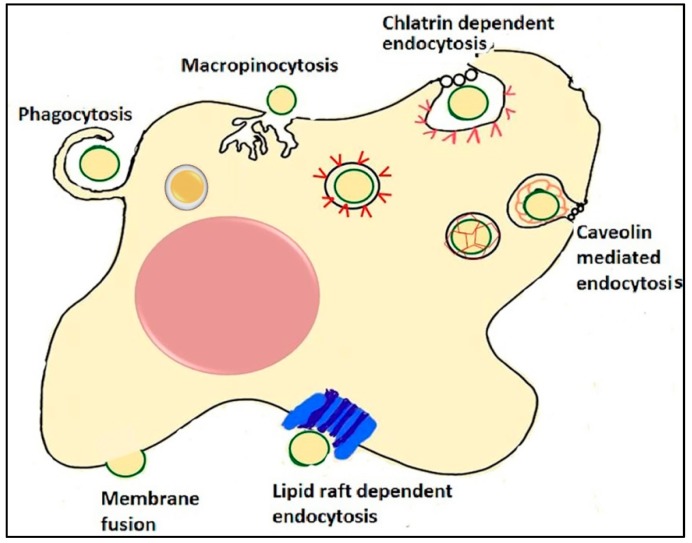
Mechanisms of EV uptake. EVs can be internalized by cells through different endocytic processes. During phagocytosis, the cytoplasm creates invaginations around the EV, which is then internalized creating an endosome. During macropinocytosis, plasma membrane ruffles fold back on themselves around the EVs, forming a lumen of a macropinosome. Clathrin-dependent endocytosis induces membrane curvature around the EV. In caveolin-mediated endocytosis caveolae (small, cave-like invaginations in the cell membrane) with EVs inside are internalized into the cell. Intracellularly, they develop chlatrin- or caveolin-coated vesicles, fuse with endosomes and deliver the cargo. EV uptake can occur by interaction of EVs with lipid rafts. Lipid rafts are involved in both clathrin- and caveolin-mediated endocytosis. Another possible way for EVs to deliver their cargo to recipient cell is by passive fusion with the plasma membrane.

**Figure 3 ijms-20-05527-f003:**
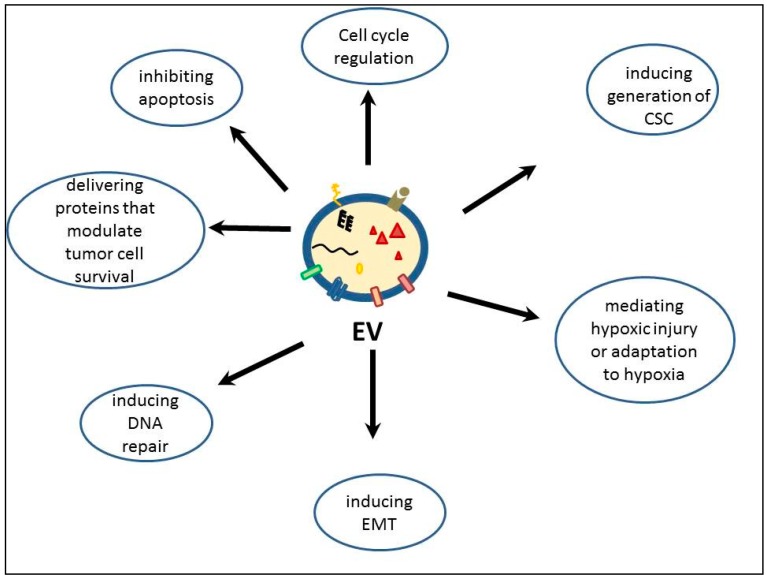
Functions of extracellular vesicles with a role in modulating radiation response. EV-extracellular vesicle; CSC-cancer stem cell; EMT-epithelial-mesenchymal transition.

**Figure 4 ijms-20-05527-f004:**
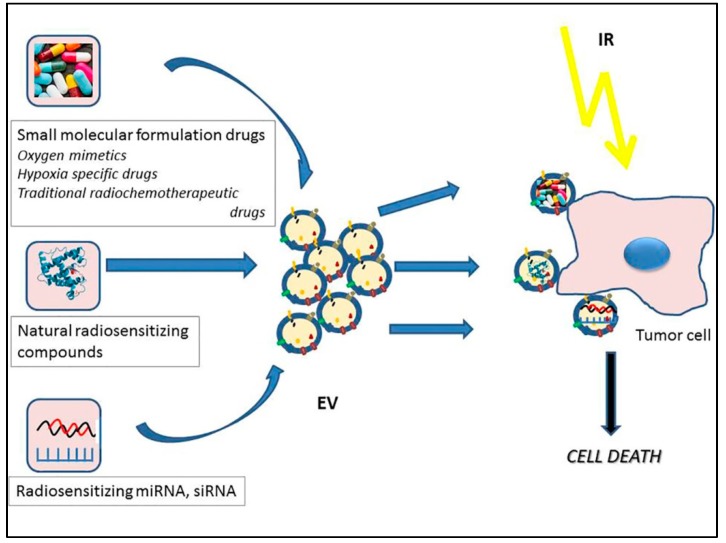
Schematic diagram depicting the potential roles of extracellular vesicles as nanocarriers for radiosensitizing agents.

**Table 1 ijms-20-05527-t001:** Examples of nanocarriers with their advantages and limitations.

Type of Nanocarrier	Advantage	Disadvantage
Liposomes [156,167]	Biocompatibility; can be loaded with both hydrophobic and hydrophilic compounds; low toxicity; can easily fuse with cell membrane	Lack of long-term stability and ability to evade the host immune system
Polymeric nanoparticles [156,167,168]	Biocompatibility and biodegradability; higher stability; targeted drug delivery; nonimmunogenicity; low toxicity	Toxic degradation, toxic monomers aggregation; difficult to scale-up
Polymeric micelles [168]	Controlled drug release; increased solubility of lipophilic compounds	Low loading capacity; usable just for lipophilic drugs
Carbon nanotubes [169]	Ease of cellular uptake; high drug loading capacity; biocompatibility; specificity to cells,	High toxicity, difficult to degrade
Gold nanoparticles [170]	Can be prepared in broad range of sizes, are easy to modify	Biocompatibility and toxicity issues
Solid lipid nanoparticles [167]	Low cost; easy to scale-up; good physical stability; good tolerability	Low drug loading; low controlability of drug release
Dendrimers [168]	Increased solubility of lipophilic compounds	Toxicity; high cost of synthesis
Extracellular vesicles	Natural origin, biocompatibility, high stability, low toxicity, capacity to evade immune degradation, possible targeted delivery	Presence of own cargo with possible diverse effects, lack of standardized isolation and loading methods

**Table 2 ijms-20-05527-t002:** Detection methods of EVs with their advantages and limitations.

Detection Methods	Principles of Detection	Advantages	Limitations
**Dynamic light scattering [173]**	Measuring EV size distribution	Accurate, reliable, and repeatable particle size analysis in very short time; Size measurement of molecules with MW < 1000Da; very low sample volume	Low refractive index of vesicles makes problematic to distinguish MVs from polydispersed and size heterogeneous samples
**Nanoparticle Tracking Analysis [174]**	Quantification of nanoscale particles in liquid suspension moving under Brownian motion	Detection of single vesicles with a diameter ≤50nm	Only semi-quantification; Inaccurate with size heterogeneous samples and particle aggregates; Considerable intra-assay count variability
**Electron microscopy**	Measuring the size and morphology of EVs	Direct assessment of morphology and size; small sample amount	Time consuming; size and morphology modifications during sample preparation
**Flow cytometry [175,176]**	EV characterization with fluorescent antibodies EV counting	Quantitative and qualitative characterization of EVs using specific markers	Detection limit of flow cytometers (>100 nm, Nonspecific: swarming effect, detection of protein/antibody aggregates
**ELISA/ Western Blot [177]**	EV characterization and quantification using specific antibodies	Standard immunological methods; specific characterization of EV protein markers	Time consuming; possible detection of non-EV proteins; nonspecific information on EV concentration/size/distribution
**Surface plasmon resonance [178]**	Label-free detection of ligand binding to target receptors immobilized on a sensing surface	Measures the total mass of EVs, including proteins, lipids, and nucleotides; small sample volumes	Inadequate quality control and normalization across study groups;
**Atomic force microscopy [179]**	EV three-dimensional topography	Fast; small sample amount	Size and morphology modifications due to sample dehydration

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
