# Peer review of "Extracellular Vesicles in Modifying the Effects of Ionizing Radiation"

_ijms, 2019, doi:10.3390/ijms20225527_

Round 1

Reviewer 1 Report

In this manuscript, the authors summarized and reviewed the studies on biological characteristics of extracellular vesicles (EVs), modification of EVs as nanocarriers in ionizing radiation, and potential benefits to improve therapeutic strategies. This manuscript is very informative and plentiful, however, somewhere with a lengthy explanation. Specific comments to further improve the manuscript are as follows.

Cells appear to take up EVs by a variety of endocytic pathways.  It would be much clearer that the authors summary Mechanisms of EV uptake by a flow diagram figure.

It is more concisely presented by some tables in the 6th part (Potential role of EVs as drug delivery nanovesicles in radiotherapy), (e.g. miRNAs and tumor types).

Authors provide an excellent explanation for relationships miRNAs and siRNAs that would affect radiosensitivity, there is an obvious lack of discussion with respect to the topic of EV delivery as radiosensitizers. Should make the brevity of that point. 

A recent paper (Mathieu, M., et al. 2019 Nat Cell Biol 21(1): 9-17) discussed the specificities of exosomes and other types of extracellular vesicles, and their roles as important agents of cell-to-cell communication. The authors should cite this paper in the manuscript.

Author Response

We are very grateful for the referee’ s nice and constructive comments on our manuscript. Below we outline our responses and the modifications made to the manuscript. With these modifications,we hope that the manuscript is now acceptable for publication.

In this manuscript, the authors summarized and reviewed the studies on biological characteristics of extracellular vesicles (EVs), modification of EVs as nanocarriers in ionizing radiation, and potential benefits to improve therapeutic strategies. This manuscript is very informative and plentiful, however, somewhere with a lengthy explanation. Specific comments to further improve the manuscript are as follows.

Cells appear to take up EVs by a variety of endocytic pathways.  It would be much clearer that the authors summary Mechanisms of EV uptake by a flow diagram figure.

Response: We have added a figure showing  mechanisms of EV uptake (Figure 2).

It is more concisely presented by some tables in the 6th part (Potential role of EVs as drug delivery nanovesicles in radiotherapy), (e.g. miRNAs and tumor types).

Response: We have added a a  figure showing the potential roles of EVs as nanocarriers  for radiosensitizing agents (figure 4).

Authors provide an excellent explanation for relationships miRNAs and siRNAs that would affect radiosensitivity, there is an obvious lack of discussion with respect to the topic of EV delivery as radiosensitizers. Should make the brevity of that point. 

Response: The topic of EVs as radiosensitizers is discussed now at lines 353-361and also was mentioned in conclusions, in lines 635-643

A recent paper (Mathieu, M., et al. 2019 Nat Cell Biol 21(1): 9-17) discussed the specificities of exosomes and other types of extracellular vesicles, and their roles as important agents of cell-to-cell communication. The authors should cite this paper in the manuscript.

Response: We introduced this citation, at lines 99 and 200 (ref 37).

Reviewer 2 Report

The review is very well written and covers all aspects about extracellular vesicles and their use supporting irradiation.

However, the review must contain some figures to support the text. For example, one figure for the EVs in general, one for uptake of EVs, and one for its proposed role as nanocarriers. For sure, other figures might also be possible.

In chapter 4 a table comparing the different types of nanocarriers with their advantages and disadvantages would be also very helpful.

Minor:

in line 82 multivesicular bodies are abbreviated by MVB. But in the following only the abbreviated MV (i. e. line 88) is used.

In line 283ff the effect of radiation in hypoxic cells is described. Here, the terms oxygen enhancement ratio and oxygen fixation hypothesis should be mentioned.

Author Response

We are very grateful for the referee’s nice and constructive comments on our manuscript. Below we outline our responses and the modifications made to the manuscript. With these modifications,we hope that the manuscript is now acceptable for publication.

The review is very well written and covers all aspects about extracellular vesicles and their use supporting irradiation.

However, the review must contain some figures to support the text. For example, one figure for the EVs in general, one for uptake of EVs, and one for its proposed role as nanocarriers. For sure, other figures might also be possible.

Response: We added these figures : Figure 1 showing EV types and composition, Figure 2 showing mechanisms of EV uptake, and Figure 4 showing roles of EVs as nanocarriers in radiotherapy.

In chapter 4 a table comparing the different types of nanocarriers with their advantages and disadvantages would be also very helpful.

Response: We have added the table (Table 1).

Minor:

in line 82 multivesicular bodies are abbreviated by MVB. But in the following only the abbreviated MV (i. e. line 88) is used.

Response: Thank you for the observation, we corrected the mistakes in the manuscript: MV stands for microvesicles( abbreviation was missing) and MVB for multivesicular bodies.

In line 283ff the effect of radiation in hypoxic cells is described. Here, the terms oxygen enhancement ratio and oxygen fixation hypothesis should be mentioned

Response: We mentioned these terms in lines 296-302.

Reviewer 3 Report

the manuscript of Szatmári et al. is summarizing the biology of extracellular vesicles and their role for radiosensitivity precisely and very well structured. I fully recommend this review for publication in MDPI and would only recommend to include a illustrative figure about EV function and cellular pathways as well as a summarizing table about detection methods, limitations and validatated in vitro/in vivo.

Author Response

We are very grateful for the referee’s nice and constructive comments on our manuscript. Below we outline our responses and the modifications made to the manuscript. With these modifications,we hope that the manuscript is now acceptable for publication:

The manuscript of Szatmári et al. is summarizing the biology of extracellular vesicles and their role for radiosensitivity precisely and very well structured. I fully recommend this review for publication in MDPI and would only recommend to include a illustrative figure about EV function and cellular pathways as well as a summarizing table about detection methods, limitations and validatated in vitro/in vivo.

Response: We introduced figure 3 for functions of EVs with a role in modulating radiation response and Table 2 and corresponding text (row 406) for EV detection methods.

Round 2

Reviewer 1 Report

The revised manuscript has reached my required modifications. Please just check the spelling and text editing. 

Reviewer 2 Report

The added figures and tables are very informative and augment the qualitiy of the review.